# Cervical Fluids Are a Source of Protein Biomarkers for Early, Non-Invasive Endometrial Cancer Diagnosis

**DOI:** 10.3390/cancers15030911

**Published:** 2023-01-31

**Authors:** Elena Martinez-Garcia, Eva Coll-de la Rubia, Antoine Lesur, Gunnar Dittmar, Antonio Gil-Moreno, Silvia Cabrera, Eva Colas

**Affiliations:** 1Proteomics of Cellular Signaling, Department of Infection and Immunity, Luxembourg Institute of Health, 1445 Strassen, Luxembourg; 2Biomedical Research Group in Gynecology, Vall Hebron Institute of Research, Universitat Autònoma de Barcelona, CIBERONC, 08035 Barcelona, Spain; 3Quantitative Biology Unit, Luxembourg Institute of Health, 1445 Strassen, Luxembourg; 4Department of Life Sciences and Medicine, University of Luxembourg, 4367 Esch-sur-Alzette, Luxembourg; 5Gynaecological Department, Vall Hebron University Hospital, CIBERONC, 08035 Barcelona, Spain

**Keywords:** Carcinoma of the endometrium, endometrial cancer, uterine cancer, biomarker, diagnosis, gynecology, endometrial sampling, proteomics, cervical sample, protein, non-invasive

## Abstract

**Simple Summary:**

Abnormal uterine bleeding (AUB) is the main symptom of endometrial cancer (EC), but it is highly nonspecific. This represents a huge burden for women’s health, since all women presenting with bleeding will undergo sequential invasive tests, avoidable in 90–95% of those women who do not have EC. This study aimed to evaluate the potential of cervical samples collected with five different devices as a source of EC diagnostic protein biomarkers. Samples collected with a Rovers Cervex Brush^®^ and the HC2 DNA collection device, Digene, were the most suitable for EC proteomic studies. A clinical retrospective study assessing the expression of 52 EC-related proteins in 41 patients (22 EC; 19 non-EC), by targeted proteomics, identified SERPINH1, VIM, TAGLN, PPIA, CSE1L, and CTNNB1 as potential EC protein biomarkers in cervical fluids (AUC > 0.8). This study opens an avenue for developing non-invasive protein-based EC diagnostic tests, which will improve the standard of care for gynecological patients.

**Abstract:**

Background: Abnormal uterine bleeding is the main symptom of endometrial cancer (EC), but it is highly nonspecific. This represents a huge burden for women’s health since all women presenting with bleeding will undergo sequential invasive tests, which are avoidable for 90–95% of those women who do not have EC. Methods: This study aimed to evaluate the potential of cervical samples collected with five different devices as a source of protein biomarkers to diagnose EC. We evaluated the protein quantity and the proteome composition of five cervical sampling methods. Results: Samples collected with a Rovers Cervex Brush^®^ and the HC2 DNA collection device, Digene, were the most suitable samples for EC proteomic studies. Most proteins found in uterine fluids were also detected in both cervical samples. We then conducted a clinical retrospective study to assess the expression of 52 EC-related proteins in 41 patients (22 EC; 19 non-EC), using targeted proteomics. We identified SERPINH1, VIM, TAGLN, PPIA, CSE1L, and CTNNB1 as potential protein biomarkers to discriminate between EC and symptomatic non-EC women with abnormal uterine bleeding in cervical fluids (AUC > 0.8). Conclusions: This study opens an avenue for developing non-invasive protein-based EC diagnostic tests, which will improve the standard of care for gynecological patients.

## 1. Introduction

Endometrial cancer (EC) is the most common gynecological disease in developed countries and its incidence and mortality rates increase yearly in tandem with the rising prevalence of obesity. Early detection is crucial for good survival outcomes. While the 5-year survival rate is 95% for local disease, it decreases dramatically to 69% in cases of regional metastasis and to 16% in cases of distant metastasis [1]. Early detection of EC relies upon the appearance of symptoms. The main symptom reported for EC is abnormal uterine bleeding (AUB), present in 90% of women diagnosed with EC. However, it is a highly nonspecific symptom, and only 5–10% of women with AUB will be eventually diagnosed with EC [2]. This represents a considerable burden for women’s health and the healthcare systems, as most women with AUB will need to undergo several invasive tests to rule out the presence of EC, but these tests would be avoidable for 90–95% of those women. The gold standard for EC diagnosis is the histopathological examination of an endometrial biopsy. The less invasive way of obtaining this biopsy is by aspiration from the uterine cavity (i.e., pipelle biopsy). This is a highly accurate diagnosis when there are cells to be analyzed in pipelle biopsies. Unfortunately, it is reported that up to 42% of pipelle biopsies either were histologically inadequate specimens, or were not performed due to technical failures [3]. In these cases, a more invasive endometrial sampling must be performed, either by hysteroscopy or dilation and curettage (D&C), to achieve a final diagnosis.

In this context, the identification of screening and diagnostic biomarkers, particularly to rule out most non-EC women, would significantly improve the diagnostic process of EC. Multiple efforts to elucidate EC biomarkers have been undertaken, mainly in blood/serum using different omics approaches [4,5,6]. Among those, HE4 and CA125 are the most studied in blood/serum. Still, their diagnostic potential, alone or in combination, remains unclear and continues to be investigated [7,8,9,10]. We previously showed the usefulness of the uterine content, both fluid and cells contained in pipelle biopsies, as a source of highly accurate protein and transcriptomic biomarkers to improve EC diagnosis [11,12,13,14]. Given the anatomical continuity of the uterine cavity with the lower genital tract, exploring cervical samples is the next step to deciphering their potential to develop non-invasive EC diagnostic tests.

Here, we evaluate the potential of exo-and/or endo-cervical samples collected with five different sampling methods to identify protein biomarkers for EC detection. We first evaluated the protein quantity and the proteome of the fluids of the different cervical samples and compared them with the proteome of uterine fluids. The two best-performing cervical sampling methods were further investigated using targeted proteomics. We measured the expression of 52 EC-related proteins in 22 EC and 19 non-EC patients and identified a set of proteins with great potential to discriminate between EC and non-EC women with AUB in these non-invasive cervical samples.

## 2. Materials and Methods

### 2.1. Patient Recruitment and Sample Collection

This study was approved by the Vall d’Hebron Hospital Ethical Committee (reference: PRAMI276-2018). The 45 patients eligible and recruited for the study were women who gave written informed consent to participate, and older than 18 years. The selected women were attending the Gynecology Department in the Vall d’Hebron Hospital to enter the EC diagnostic process due to the presence of common symptoms of EC, including AUB. Exclusion criteria were women with an active pelvic infection, and women who have had viral infections with evidence of active and latent disease, such as Hepatitis B, C, and HIV infection. All samples were collected between June 2016 and January 2017. Five different exo- and/or endo-cervical samples and one pipelle biopsy were collected from four women (two EC and two non-EC) to assess the feasibility of measuring proteins in cervical samples, and the two best performing cervical samples and a pipelle biopsy were collected from 41 women (22 EC and 19 non-EC) for the verification of EC biomarkers. The clinicopathological characteristics of all the patients included in this study are shown in Table 1. In all women, diagnosis was confirmed by the histopathological examination of the resected tissue obtained during the surgical treatment (for EC patients), or the histopathological examination of an endometrial biopsy (for non-EC patients). This information was used as the reference standard for the study.

For each patient, cervical samples were obtained first (M1 > M2 > M3 > M4 > M5), followed by pipelle biopsy. The five different cervical samples (M1–M5) were collected with five different brushes, and dipped into falcon tubes containing different volumes of 1X PBS (Figure 1A–C):
M1 was obtained with the Rovers Cervex Brush^®^ (Rovers Medical Devices, Oss, The Netherlands). This cervical brush is used to obtain samples for cervical liquid cytology. It has a shape designed to obtain a good representation of endocervical and exocervical material.M2 was obtained with the Wooden cervical scrape or Ayres spatula (Goodwood medical care, Dalian, China). It is generally used to obtain the exocervical representation of the pap-smears. It can also be used to obtain a vaginal sample but it was not used for this purpose in the present study.M3 was obtained with the endocervical swab HC2 DNA collection device Digene (QIAGEN, Hilden, Germany), used to get an endocervical mucus sample. It is generally used to perform the hybrid capture test to rule out human papillomavirus infections of the lower genital tract. It has little bristles that are introduced in the endocervix to collect the mucus.M4 was obtained with a cotton swab (Deltalab, Barcelona, Spain), usually used to take superficial samples to perform bacterial cultures of any location. It is blunt and only impregnates with the secretions of the endocervix without scratching the tissue.M5 was obtained using an endocervical brush (Bexen medical, Mondragón, Spain). It is the tool used to obtain an endocervical representation in pap-smears.Pipelle biopsies were obtained with the Cornier Pipelle (Eurogine Ref. 03040200, Spain) following the standard procedure. The device was introduced through the cervical canal into the uterine cavity, and the pipelle biopsy was obtained by applying negative pressure. This device requires cervical permeability to access the uterine cavity. Once collected, PBS 1X was added in a 1:1 volume.

### 2.2. Sample Preparation

All cervical and uterine samples in PBS were centrifuged at 2500× *g* for 20 min to separate the fluid from the cellular fraction. Uterine fluids (UF) obtained from pipelle biopsies and cervical fluids (CF) were stored at −80 °C until used. Whilst UF were analyzed without and with depletion of albumin and IgGs (raw and depleted respectively), CF were not depleted. For depletion, UF samples were sonicated (Labsonic M, Sartorius Stedim Biotech, Göttingen, Germany) during 5 cycles of 5 s, and 50 µL of each sample was processed with the Albumin & IgG depletion spin trap kit (GE Healthcare, Chicago, IL, USA) according to the manufacturer’s instructions.

The total protein concentration of all samples was measured by the Bradford assay. Equal amounts of protein per sample were used for the proteomic analysis. First, samples were denatured by addition of urea to a final concentration of 6 M, incubated at 22 °C under agitation for 20 min, followed by a 10 min incubation in an ultrasonic bath (Branson 5510, Branson Ultrasonics, Brookfield, WI, USA). The proteins were then reduced in 5 mM dithiothreitol (DTT) for 1 h at 37 °C, and alkylated in 15 mM iodoacetamide (IAA) for 30 min at 22 °C in the dark. Each sample was then digested, first with LysC protease (Wako, Richmond, VA, USA) at a protease/total protein amount ratio of 1/50 (*w*/*w*) for 4 h at 37 °C, and then with trypsin (Promega, Madison, WI, USA) overnight at 37 °C at an enzyme/substrate ratio of 1/25 (*w*/*w*), after dilution of urea to a final concentration of 1 M. Proteolysis was stopped by addition of 1 µL of formic acid per 100 µL of solution. For the LC-PRM studies, samples were digested with a 1/150 (*w*/*w*) ratio of LysC and 1/50 (*w*/*w*) ratio of trypsin, proteolysis was stopped with formic acid, and a mixture of stable isotope labeled synthetic peptides (Thermo Fisher Scientific, Waltham, MA, USA, crude quality) was spiked in each sample (C terminal arginine 13C6, 15N4, C terminal lysine 13C6, 15N2, or when it was not applicable with a heavy leucine 13C6, 15N1, or phenylalanine 13C9, 15N1). All samples were desalted onto solid phase extraction cartridges (Sep Pak tC18, 25 mg, Waters, Milford, CT, USA) and dried using a vacuum centrifuge.

### 2.3. DDA Analysis on a Tims-TOF Pro Mass Spectrometer

Seven different matrices corresponding to 5 CF, and raw UF and depleted UF from four different patients (n = 28 samples), were analyzed by nano-UHPLC (nanoElute, Bruker Daltonics, Billerica, MA, USA) coupled to a tims-TOF pro mass spectrometer. The samples were directly injected onto a reverse phase column (250 mm × 75 µm, 1.6 µm, C18; IonOptiks, Fitzroy, VIC, Australia) heated at 50 °C. The mobile phases consisted of 0.1% (*v*/*v*) formic acid in water (phase A) and in acetonitrile (phase B). The samples were separated by a 100 min stepped gradient ranging from 2–30% B at a flow rate of 400 nL/min. The nano-UHPLC was coupled with a tims-TOF pro instrument (Bruker Daltonics) operated in dda-PASEF mode. Survey scans were acquired from 100 to 1700 *m*/*z* within an ion moblity range of 0.6 to 1.6 s/cm^2^. The ion mobility accumulation and separation time were both set to 100 ms. The mass spectrometry (MS)/MS acquisition scheme was set to PASEF mode, with 10 PASEF events per MS cycle (total cycle time 1.15 s). Dynamic exclusion of fragmented ion precursors was set to 0.4 min.

### 2.4. DDA Data Processing

MS files were analyzed in the MaxQuant software, version 2.0.1.0 [15]. The MS/MS spectra were searched by the Andromeda search engine against the TrEMBL UniProt Homo sapiens (August 2021, 78,139 entries). Trypsin/P was specified as the protease and a maximum of three missed cleavages was allowed. Carbamidomethyl (C) was set as a fixed modification and acetyl (protein N-terminus) and oxidation (M) were set as variable modifications. The ”match between runs” option was enabled. The false discovery rate (FDR) for peptide and protein identifications was set to 1%. For protein quantification, label-free quantification (LFQ) was performed, with a minimum ratio count of 2 [16].

Data analysis was performed in the Perseus platform, version 1.6.12.0 [17]. The raw data was filtered for “contaminants” and “reverse” identifications, log2-transformed, and further filtered for 70% data completeness across all samples. The missing values were imputed from a normal distribution (downshifted mean by 1.8 standard deviation (SD) and scaled SD (0.3) relative to that of the proteome abundance distribution). This dataset was used for hierarchical clustering after z-score normalization, and to build multi scatter plots to evaluate the correlation between the uterine and cervical samples.

### 2.5. PRM Analysis

The separation of the peptides was performed on a Dionex Ultimate 3000 RSLC chromatography system operated in column-switching mode. The equivalent of 250 ng of digested sample was injected and loaded onto a trap column (75 µm × 2 cm, C18 pepmap 100, 3 µm) using a mobile phase of 0.05% trifluoroacetic acid and 1% acetonitrile in water at a flow rate of 5 µL/min. Peptides were further eluted onto the analytical column (75 µm × 15 cm, C18 pepmap 100, 2 µm) at 300 nL/min by a linear gradient starting from 2% solvent A to 35% solvent B in 48 min. The solvent A was 0.1% formic acid in water and the solvent B was 0.1% formic acid in acetonitrile.

The PRM analysis was performed on a Q Exactive HF mass spectrometer (Thermo Fisher Scientific). The MS cycle consisted of a full MS1 scan performed at a resolving power of 60,000 (at 200 *m*/*z*), followed by time scheduled targeted MS2 scans, with a normalized collision energy of 20, acquired at a resolving power of 60,000 (at 200 *m*/*z*), maximum accumulation time of 110 ms, and an AGC target of 1e6 charges. The quadrupole isolation window of precursor ions was set to 1 *m*/*z* unit for the MS2 events, and the duration of the time scheduled windows for each pair of endogenous and isotopically labeled peptides was set to 2 min.

### 2.6. Data Analysis

The statistical analysis was performed in SPSS (v20.0) (IBM, Armonk, NY, USA) and Graph Pad Prism (v.6.0) (GraphPad Software, La Jolla, CA, USA). The Pearson correlation of the expression levels of the peptides belonging to one protein was calculated. Statistical analysis to determine the diagnostic performance of each protein was performed with one peptide if the correlation between peptides was higher than 0.80, or with two peptides treated as independent entities if the correlation was lower than 0.80. Comparison of the levels of the targeted peptides between EC and non-EC patients was performed using the non-parametric Mann–Whitney U test. *p*-values were adjusted for multiple comparisons using the Benjamini–Hochberg FDR method [18]. An FDR lower than 0.05 was considered statistically significant. ROC analysis was used to assess the specificity and sensitivity of the biomarkers, and the area under the ROC curve (AUC) was estimated for each individual protein. A Pearson correlation was used to compare the levels of expression of the different proteins between matrices.

## 3. Results

### 3.1. Proteomic Characterization of Cervical Fluids

Cervical samples were collected using different commercially available sampling devices that differed in their shape and utility. M2 was used for the collection of exocervical samples; M3, M4, and M5 for the collection of endocervical samples; and M1 allowed for the collection of both endo- and exocervical samples (Figure 1A,B). All cervical samples were diluted in a standard saline solution, i.e., PBS, and centrifuged in order to obtain the CF (Figure 1C). The workflow followed in this study to investigate the suitability of those fluids for proteomic studies, and to identify EC protein biomarkers, is depicted in Figure 1D.

Firstly, we evaluated whether the protein concentration of the different CF was sufficient for proteomic analysis by MS, as all cervical samples were highly diluted in PBS to recover the samples out of the brushes. We measured the total protein concentration of the five CF and the uterine fluid before (raw) and after depletion of albumin and IgGs (depleted) from four different women (two EC and two non-EC). As expected, raw UF had a much higher protein concentration (average of 32 µg/µL) than any other sample since they contained highly abundant blood proteins (Figure 2A). This protein concentration dropped to an average of 0.95 µg/µL when albumin and IgGs were depleted from the UF, and a similar protein concentration was quantified in the CFs. M1, M3, and M5 samples showed the highest protein concentration (averages of 1.95 µg/µL, 1.86 µg/µL, 1.45 µg/µL, respectively), which was sufficient for further proteomic analysis. On the contrary, M2 and M4 yielded a low protein concentration, which could hamper their use for MS analysis (Figure 2A).

The number of protein identifications was quite similar among all samples, ranging from an average of 1600 proteins detected in M1 and M2 samples, to an average of about 1800 proteins detected in UF, M3, M4, and M5 samples (Figure 2B). UF samples were included in this study as they have been shown to be a good source of highly accurate EC diagnostic biomarkers [13], and are in direct contact with the tumor in the endometrium. The five different CF samples shared a higher number of proteins among them (84–90.7% proteins in common) than with the UF (79–86%), with M1 being the most different CF sample. Still, 97.6% of the proteins identified in CF samples were also identified in raw or depleted UF samples, indicating that the CF share to a great extent the proteome of UF (Figure 2C,D).

Next, we assessed the detectability of EC-related proteins in CF. A list of 506 proteins described in the literature as potential EC diagnostic biomarkers, mostly derived from studies performed on endometrial tissue samples, were considered as EC-related proteins [13]. From those, 171 proteins were identified in any of the fluids analyzed (Figure 2E), and 153 of them (89%) were detected in all the analyzed matrices. Interestingly, 14 of the 20 most validated EC biomarkers in tissue samples [13] were detected in all the different CF samples, including the two most studied EC diagnostic biomarkers (HE4 and CA125).

Ultimately, we compared the protein abundance in the uterine and cervical fluid proteome. As shown in Figure 2F, the UF and CF samples formed two different clusters for the four analyzed patients, except for one non-EC patient (Patient 3), suggesting that the protein levels in UF were different to those in the cervical samples. The correlation of protein abundance between raw and depleted UF was very high (r = 0.97–1). Importantly, this indicates that the depletion of albumin and IgGs adds no significant improvement and could be avoided. The correlation in protein abundance was also high among the different CF (r = 0.81–0.90). The three samples collected at the endocervix, M3, M4, and M5, clustered together in the heatmap and showed the highest similarity compared to M1 and M2, which were samples also representing the exocervix. The correlation between UF and CF was significantly lower (r = 0.6). This indicates that, although most of the proteins found in the UF can be found in the cervical samples, the levels of those proteins might differ between sites.

### 3.2. Measurement of EC-Related Protein Biomarkers in CF

In Martinez-Garcia et al. [13,14], 52 proteins were studied as potential EC diagnostic biomarkers in the depleted uterine fluids of 116 patients. In the present study, we assessed the detection of those 52 proteins in the five different CF and raw UF samples from four patients (two EC; two non-EC), measuring 98 peptides by targeted proteomics (Appendix A). Most of these 52 proteins were detected in the raw UF sample and in the CF samples, with the highest number of biomarkers detected in M1 (51 proteins) and M3 (50 proteins) (Figure 3).

Among the 52 proteins, 8 proteins previously achieved the highest accuracy to diagnose EC in depleted UF (AUC = 0.85–0.91): MMP9, KPYM, LDHA, CADH1, NAMPT, PERM, ENOA, and CAPG [13,14]. Here, the highest correlation between the levels of these 8 proteins in the CF and depleted UF samples was observed for M1 (r = 0.73–0.98) and M3 (r = 0.74–0.94, except for MMP9 and PERM), while the other cervical samples showed no correlation (Figure 3B). Despite the limited number of patients, we observed that these 8 protein biomarkers were significantly higher in EC compared to non-EC patients not only in the depleted UF samples, but also when measured in the raw UF samples and in the M1 and M3 CF samples, with fold changes > 2 for the 8 proteins (Figure 3C).

### 3.3. Selection of M1 and M3 Cervical Samples

Among the five different CF samples analyzed, the one collected with the Rovers Cervex Brush^®^ (M1) and the one collected with the endocervical swab HC2 DNA collection device, Digene^®^, (M3) were selected as the most suitable sampling methods to subsequently perform proteomic studies to identify EC protein biomarkers. On one hand, M1, M3, and M5 yielded the highest protein concentration. On the other hand, we detected the highest number of EC biomarker candidates in M1 and M3, and we observed the highest correlation between these two samples and uterine fluids for the eight most promising EC biomarkers targeted, as well as the largest differences of the levels of those eight proteins between EC and non-EC patients.

### 3.4. Verification Study of EC Biomarker Candidates in Raw UF and CF

In order to evaluate the potential of the M1 and M3 CF samples to provide EC diagnostic biomarkers, we measured the previously selected 52 EC-related proteins in raw UF, M1 and M3 CF samples from 41 patients (22 EC and 19 non-EC women) using the PRM approach (Appendix A). From the 98 targeted peptides, four peptides were excluded as they were detected in less than 50% of the samples, leading to a total of 51 proteins robustly measured with 94 peptides. Triplicates of four pooled UF samples, four pooled M1 samples, and four pooled M3 samples were included to determine the reproducibility of the whole sample preparation and MS analysis. We obtained an average coefficient of variation (CV) of 4%, 4%, and 3% for the three pools, respectively. Only 3 out of the 282 measurements (94 peptides quantified in the three pools) showed a CV higher than 10%, and none above 20%, highlighting the robustness of the quantification in these three complex matrices (Appendix A).

The highest number of significant protein biomarkers were identified in the raw UF samples. A total of 36 proteins (63 peptides) showed significantly higher levels in EC compared to non-EC women, with an FDR < 0.05, a fold change (FC) value > 2, and an AUC higher than 0.71 (Figure 4A). Among those, LDHA, ENO1, and PKM showed the highest accuracy to discriminate between EC and non-EC patients with an AUC higher than 0.9, and specificity and sensitivity values over 85% (Figure 4B). Interestingly, 28 out of these 36 proteins were also able to differentiate between EC and non-EC patients, with AUC higher than 0.7 in M1 and/or M3 CF samples (Figure 4A). However, the best performing proteins in UF were not the best performing in CF (Appendix A).

The levels of all the proteins measured in the M1 cervical fluid samples showed a high variability among the different patients, and thus none of these proteins had an FDR < 0.05. Despite this, 15 proteins discriminated EC and non-EC patients with an AUC higher than 0.7 (Figure 5A, Appendix A). The three best performing proteins were SERPHINH1 (FC = 5.89, AUC = 0.83, 83% sensitivity and 81% specificity), VIM (FC = 2.65, AUC = 0.80, 78% sensitivity and 81% specificity), and CSE1L (FC = 2.17, AUC = 0.79, 83% sensitivity and 81% specificity) (Figure 5C).

EC biomarker candidates measured in the M3 cervical fluid samples resulted in the identification of 28 proteins differentially expressed between EC and non-EC patients (FDR < 0.05), all of them with an AUC higher than 0.7 (Figure 5B, Appendix A). The top three performing biomarkers were TAGLN (FC = 6.38, AUC = 0.84, 79% sensitivity and 72% specificity), PPIA (FC = 1.84, AUC = 0.83, 79% sensitivity and 78% specificity), and CSE1L (FC = 2.25, AUC = 0.83, 89% sensitivity and 72% specificity) (Figure 5D). Three biomarkers, VIM, LGALS1, and FSCN1, only showed diagnostic potential in CF but not in UF, highlighting that both uterine and cervical fluids can provide accurate complementary EC diagnostic biomarkers. The best performing biomarkers identified in M1 and M3 samples were able to discriminate between EC and non-EC patients independently of histological type and the grade of the tumor (Appendix A).

## 4. Discussion

Currently, EC diagnosis relies on the observation of tumor cells in endometrial biopsies, mainly obtained by aspiration (i.e., pipelle biopsies). However, this procedure fails to give a final diagnosis in up to 42% of cases, due to technical failure in obtaining an adequate sample or to the scarce cellularity in the pipelle biopsy samples [3]. Therefore, the development of a simple, non-invasive test that accurately identifies EC among symptomatic women and reassures healthy women would transform patient care. This test would also help in the screening of high-risk populations, such as women diagnosed with Lynch Syndrome, women with a high BMI, women taking Tamoxifen, and/or women who were diagnosed with a thickened endometrium, among others. In this context, the use of non-invasive samples collected in the lower genital tract (i.e., the cervix or vagina) for EC diagnosis has emerged as a promising alternative [19].

Cervical samples are well accepted by women, and they are an already established and routinely used tool for cervical cancer screening worldwide, but not for EC, due to the lack of sensitivity for the diagnosis of this disease. Many studies have evaluated the sensitivity of cervicovaginal cytology for the detection of EC. Due to the anatomical continuity of the uterine cavity with the cervix, it was proven that EC shed malignant cells that can be collected in a less invasive way in the cervix. However, only around 45% of patients with EC showed an abnormal Pap test result [20]. An interesting alternative is the assessment of tumor biomarkers. To date, most studies have focused on the evaluation of the methylation levels and/or mutational profiles of tumor DNA from samples collected in the cervix or the vagina with different devices: cervical scrapes, vaginal swabs, vaginal tampons, etc. [21,22,23,24,25]. However, the application of those results into the clinical practice is still far away, as the required techniques are currently not available in clinical laboratories. However, proteins are more stable than DNA and RNA, thus providing easy sample collection, preparation, and storage conditions. Importantly, protein levels in blood and biofluids are routinely assessed by automatized machines in a highly efficient and economic manner in all clinical laboratories. To our knowledge, only one study has characterized the proteome of cervico-vaginal samples, using a pool of nine cancer cases, three atypical hyperplasia and seven controls [26]. This study generated a spectral library to be used in subsequent biomarker research studies but did not describe any potential EC biomarker. In this study, we evaluated five different sampling methods of the exo- and endocervix to select the best method to perform proteomic biomarker studies, and elucidate potential EC diagnostic biomarkers in UF and CF samples.

Initially, we compared the proteome of CF samples collected with five different sampling methods, and also, the proteome of UF, proved to be a relevant source of EC protein biomarkers. Whilst CF shared 97% of their proteome with UF, including most of the 52 EC biomarkers described in Martinez-Garcia E et al. [13], the levels of the proteins greatly differ between these samples obtained at different sites. Indeed, CFs collected at different sites also presented slight differences. The correlation in protein abundance was high among the different CF (r = 0.81–0.90) samples, but the three samples collected at the endocervix, M3, M4, and M5, showed the highest similarity; compared to M1 and M2, which were samples also representing the exocervix. These results highlight the importance to set up the preanalytical conditions of sample collection and processing prior to initiating a biomarker research study.

The current study revealed that the most promising CF for proteomic studies were those collected with the Rovers Cervex Brush^®^ (M1) and the HC2 DNA collection device Digene (M3), both of them diluted in a saline solution. These two devices, that allow the collection of endocervical fluid, yielded the highest protein concentrations. They also allowed the identification of the highest number of protein biomarkers (M1: 51 proteins; M3: 50 proteins) among the 52 EC potential biomarkers analyzed; and finally, they showed the highest correlation with UF for the eight most promising EC biomarkers, as well as the largest differences of those eight proteins between EC and non-EC patients.

Importantly, this study evaluated the potential of M1 and M3 CF samples as a source of EC biomarkers by measuring 52 EC potential biomarker using a targeted MS-based approach in a retrospective clinical study including 41 patients. A total of 15 and 28 proteins were significantly higher in M1 and M3 samples from EC compared to non-EC patients, respectively. Importantly, SERPH and VIME were able to discriminate between EC and non-EC patients with an AUC > 0.80 when measured in the M1 samples. In M3, eight proteins achieved an AUC > 0.80, including TAGLN, PPIA, and CSE1L. These results validate the use of M1 and M3 CF samples as an untapped source of non-invasive EC protein biomarkers. Now, the diagnostic performance of the biomarkers here described needs further validation in a larger and independent cohort of patients, and the combination of several of these proteins should be evaluated to increase the diagnostic power of the individual proteins.

Although it was not the primary aim of this study, we also elucidated the potential use of raw UF for EC diagnosis. In a previous study performed by our group, we measured the levels of 52 potential protein biomarkers in UFs from two independent cohorts of 38 and 116 EC and non-EC patients, respectively. We developed a 2-protein panel able to diagnose EC with 94% sensitivity and 87% specificity, and achieving a correct diagnosis for all women that could not be diagnosed with the histopathologic examination of their pipelle biopsies [14]. UF samples used in this study were depleted of albumin and IgGs to facilitate the detection of lower abundance proteins. In the current study, we demonstrated that most of the proteins evaluated in depleted UF samples can be detected in the non-depleted UF samples, which we named raw UF. Indeed, our results show a high correlation of the proteins in raw and depleted samples, both for the whole proteome (r = 0.75–0.99) and for EC-biomarker proteins (r = 0.97–0.99). Additionally, we confirmed 36/52 proteins as biomarkers for EC diagnosis when measured in raw UF. Among the best performing biomarkers in raw UF are LDHA, ENO1, KPYM, and CASP3, with an AUC higher than 0.9. These results are promising for the implementation of these biomarkers to improve EC diagnosis, since larger prospective validation studies can be performed in UF without the need of prior depletion steps.

A limitation of this study is that we restricted the identification of EC biomarkers in CF to the already known EC biomarkers which have been described in UF. Although we revealed the potential of some of these EC biomarkers in CF, we also showed the strong differences in the protein abundance of CF and UF samples, with low correlations at the whole proteome level and when focusing on the 52 EC biomarker candidates. Consequently, we highlight the need for designing discovery studies in CF to identify additional EC biomarkers, and this could be achieved by using M1 and M3 CF samples. Another limitation is the high inter-individual variability in protein biomarker detection observed in the verification study. The main cause could be the broad biological variability of women recruited in this study, since we unrestrictedly included all women entering the diagnostic process for EC. Consequently, this potential of our biomarkers should be evaluated considering the multiple cofounding factors that exist in this population: different pathologies among non-EC patients and EC patients, histological and molecular subtypes of EC, age of the patients, BMI, ethnicity, etc. Future clinical studies should be performed to advance in this direction.

## 5. Conclusions

This study demonstrated the potential of CF as an untapped source of EC protein biomarkers. Among five different methods to collect cervical samples, we provided evidence that the Rovers Cervex Brush^®^ and the HC2 DNA collection device, Digene, are the most suitable methods to obtain a high quality and quantity protein material to perform proteomic studies on CF. Notably, we identified SERPINH1, VIM, TAGLN, PPIA, CSE1L, and CTNNB1 proteins as promising biomarkers for EC diagnosis in CF. This study opens an avenue for the development of non-invasive protein-based tools for EC diagnosis.

## Figures and Tables

**Figure 1 cancers-15-00911-f001:**
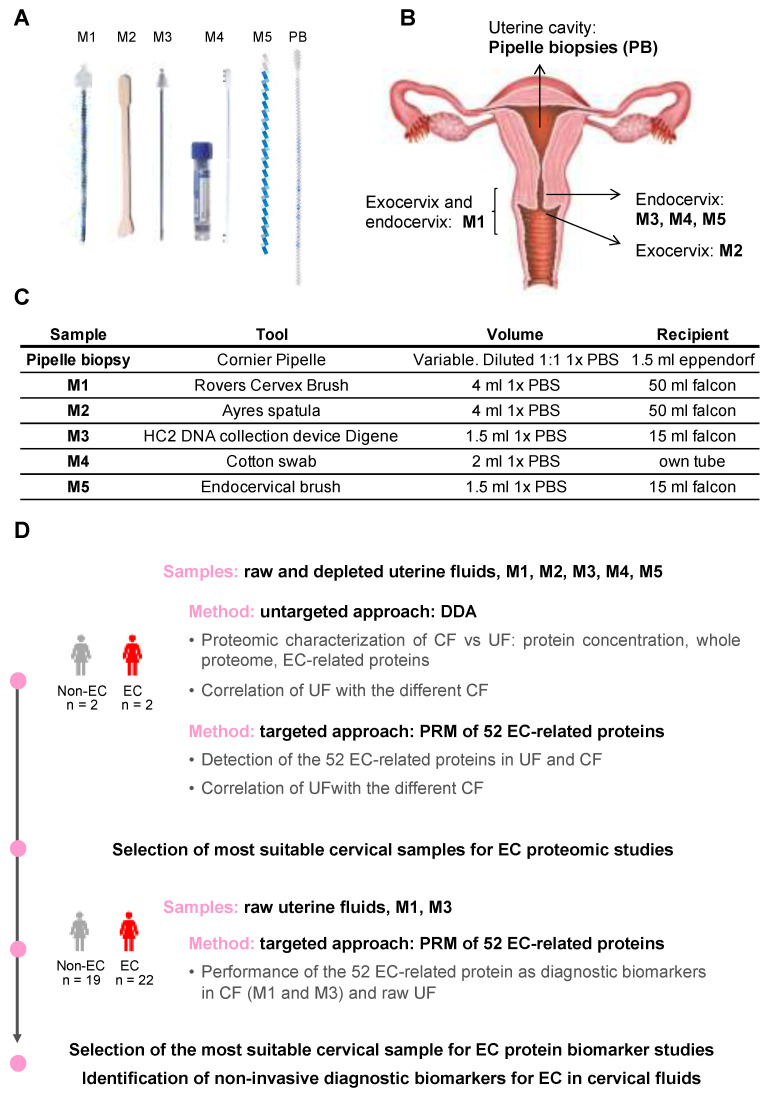
Clinical samples evaluated and general workflow of the study. (**A**): Medical tools used to collect the samples: brushes (for cervical samples M1–M5) and Cornier Pipelle (for pipelle biopsies, PB). (**B**): Collection site of the different samples. (**C**): Description of the methodology (tools, volume, and recipient) followed to collect the samples. (**D**): General workflow of the study. EC: endometrial cancer, DDA: Data Dependent Acquisition; PRM: Parallel Reaction Monitoring; UF: uterine fluid obtained from the pipelle biopsies; CF: cervical fluid.

**Figure 2 cancers-15-00911-f002:**
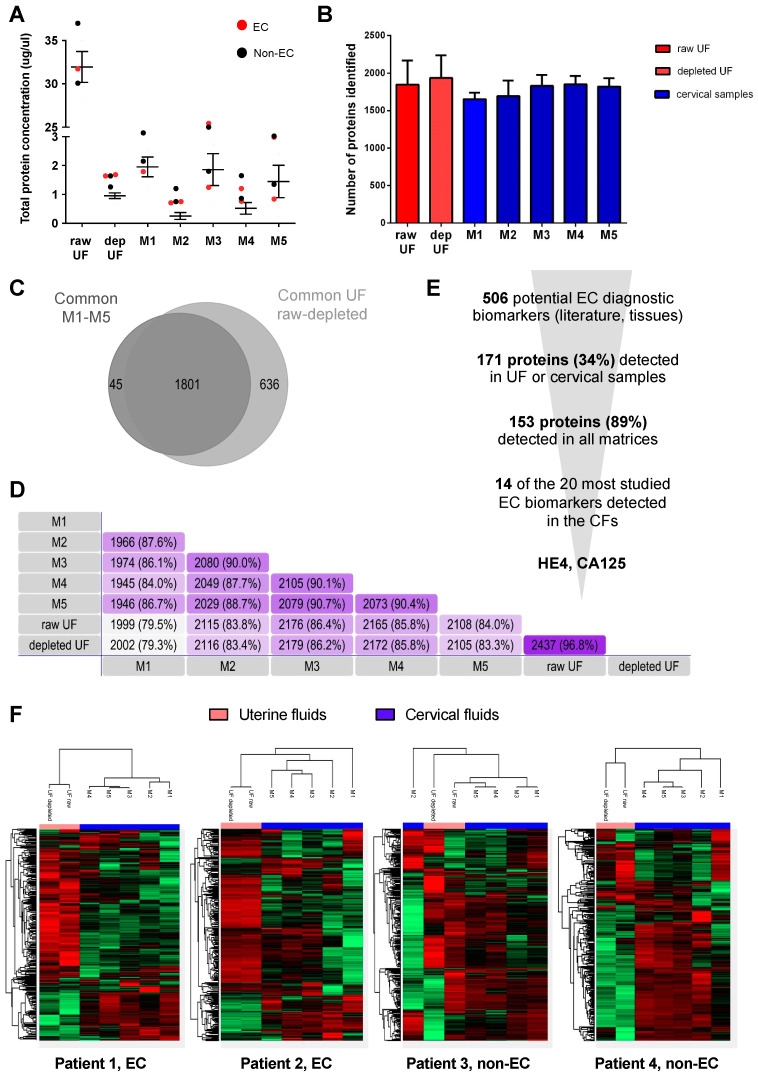
Protein concentration and proteome comparison of uterine and cervical fluids (2 EC, 2 non-EC). (**A**): Total protein concentration (µg/µL) of raw and depleted uterine fluids and cervical fluids M1–M5. (**B**): Barplot representing the total number of proteins identified by MS in the four patients for each matrix analyzed (in red uterine fluids, and in blue cervical fluids). (**C**): Venn diagram with the common and unique proteins identified in the uterine fluids versus cervical fluids. (**D**): Venn diagram showing the number of proteins in common between the different matrices. (**E**): Detectability of potential EC diagnostic biomarkers in the different matrices. (**F**): Heatmaps showing how the UFs and CFs of the four patients analyzed are clustered based on the protein levels. UF: uterine fluid; dep: depleted.

**Figure 3 cancers-15-00911-f003:**
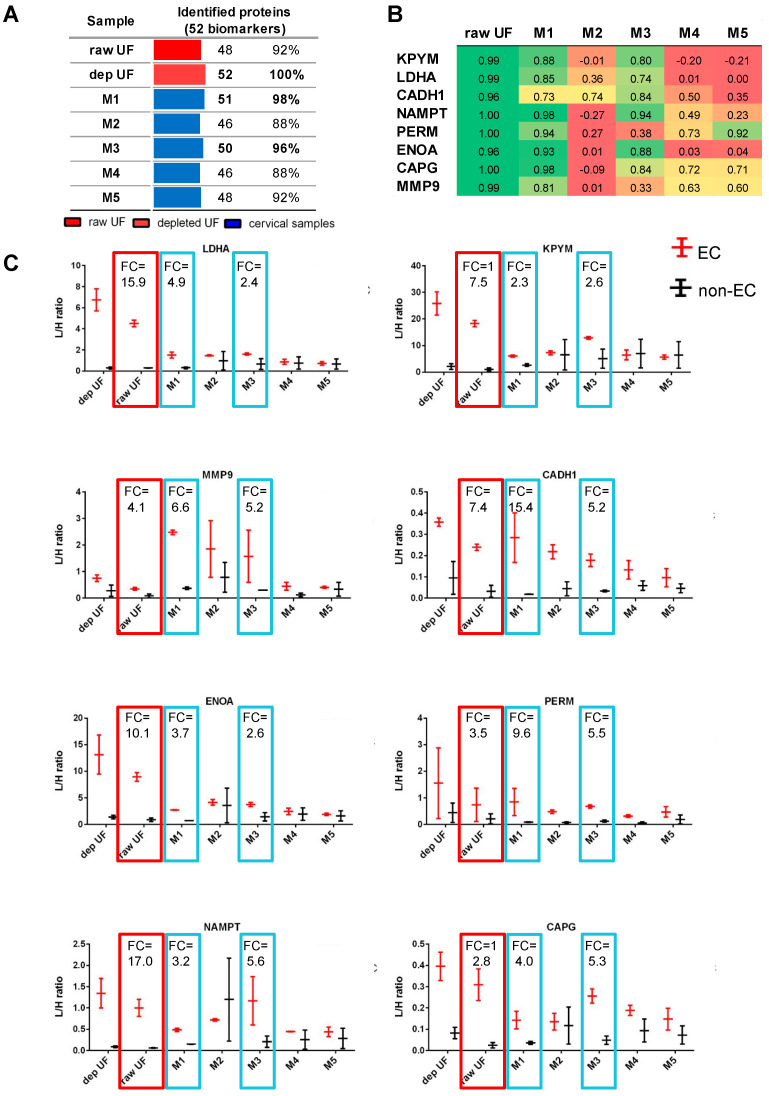
Quantification of 52 EC diagnostic biomarkers in 7 different matrices from 4 patients. (**A**): Bar plots representing the number of proteins out of the 52 EC biomarkers identified in each matrix (in red uterine fluids, and in blue cervical fluids). (**B**): Pearson correlation coefficients showing the degree of correlation between the levels of the 8 best performing EC biomarkers across the 4 patients when measured in the depleted uterine fluid versus each of the other matrices analyzed. (**C**): Dot plots representing the levels (L/H ratios) of the 8 EC protein biomarkers in the different matrices of the EC (in red) and non-EC (in black) patients. The highest fold changes between EC and non-EC patients are observed in the uterine fluids (red square), and M1 and M3 cervical samples (blue squares) for all the 8 proteins. L/H ratio: light/heavy ratios; FC: fold change.

**Figure 4 cancers-15-00911-f004:**
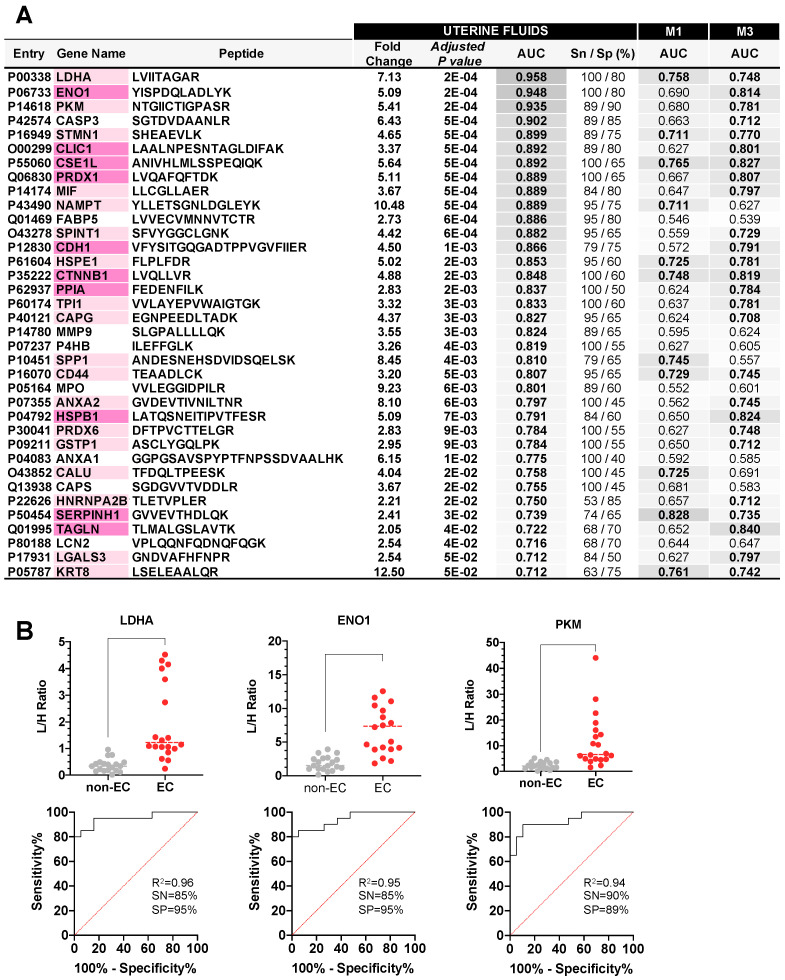
Proteins showing significantly different levels in uterine fluids from EC (n = 20) and non-EC women (n = 19). (**A**): Statistical results of the 36 proteins presenting an FDR < 0.05, Fold Change > |2| and AUC > 0.7 in the raw uterine fluids from the verification study (n = 39) when comparing EC vs. non-EC women. AUC values for the same proteins measured in M1 and M3 cervical fluids are also shown. AUC values are graded in grey according to their discriminative power. Proteins that demonstrated their diagnostic potential also in cervical fluids are highlighted in pink (AUC > 0.7), and in dark pink the ones with AUC values higher than 0.8 in cervical fluids. (**B**): Dot plots of the top three performing proteins representing the light/heavy ratios (L/H) obtained by LC-MS/MS PRM when analyzed in raw uterine fluids, and their corresponding ROC curve and sensitivity and specificity values. SN: sensitivity; SP: specificity.

**Figure 5 cancers-15-00911-f005:**
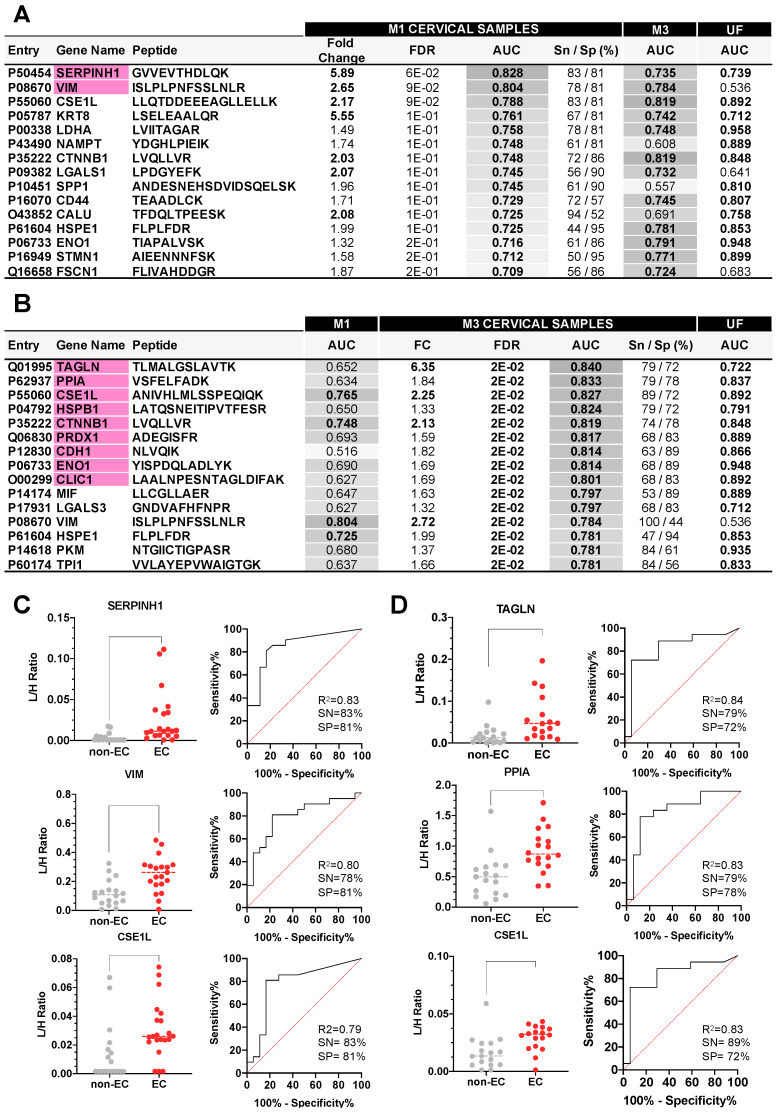
Proteins showing significantly different levels in M1 and M3 cervical fluids from 39 patients. (**A**): Statistical results of the 15 proteins presenting an AUC > 0.7 in the M1 cervical fluids from the verification study when comparing EC (n = 21) vs. non-EC (n = 18) women. Proteins with AUC values higher than 0.8 are highlighted in pink. AUC values for the same proteins measured in the M3 cervical fluids and the raw uterine fluids are also shown. These values are graded in grey according to their discriminative power. (**B**): Top 15 proteins presenting an FDR < 0.05, Fold Change > |1.3| and AUC > 0.7 in the M3 cervical fluid of 37 patients when comparing EC (n = 18) vs. non-EC women (n = 19). Proteins with AUC values higher than 0.8 are highlighted in pink. AUC values for the same proteins measured in the M1 cervical fluids and the raw uterine fluids are also shown. These values are graded in grey according to their discriminative power. (**C**): Dot plots of the top three performing proteins representing the light/heavy ratios (L/H) obtained by LC-MS/MS PRM when analyzed in M1 cervical fluids, and their corresponding ROC curve. (**D**): Dot plots of the top three performing proteins representing the light/heavy ratios (L/H) obtained by LC-MS/MS PRM when analyzed in M3 cervical fluids, and their corresponding ROC curve.

**Table 1 cancers-15-00911-t001:** Clinicopathological features of the patients included in the study. EC: endometrial cancer; depl: depleted sample from Albumin and IgGs.

		Method Optimization		Verification Phase	
		EC	non-EC	EC	non-EC
		(n = 2)	(n = 2)	(n = 22)	(n = 19)
**Age (years)**					
	Mean	69	50	70	60
	Minimum	63	48	49	23
	Maximum	75	51	93	88
**Uterine condition**					
	Premenopausal	-	1	1	3
	Postmenopausal	2	1	21	16
**Benign gynecological condition**					
	Atrophic endometrium		2		13
	Normal endometrium		-		3
	Endometrial polyp		-		2
	Simple hyperplasia, no atypia		-		1
**Histological type**					
	Endometrioid	2		18	
	Serous	-		3	
	Others (carcinosarcoma)	-		1	
**Histological grade**					
	Low-grade	1		14	
	High-grade	1		8	
**FIGO stage**					
	IA	1		9	
	IB	1		5	
	II	-		4	
	IIIC2	-		3	
	IVB	-		1	
**Miometrial invasion**					
	<50%	1		13	
	>50%	1		9	
**Lymphovascular invasion**					
	Yes	-		7	
	No	2		15	
**Molecular classification**					
	POLEmut	-		-	
	MMRd	2		9	
	NSMP	-		5	
	p53mut	-		2	
	NA	-		6	
**Samples collected**					
	Pipelle biopsy (PB)	PB		PB	
	Cervical samples	M1, M2, M3, M4, M5		M1, M3	

## Data Availability

The data presented in this study are available on request from the corresponding author.

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
