# Peer review of "Cervical Fluids Are a Source of Protein Biomarkers for Early, Non-Invasive Endometrial Cancer Diagnosis"

_cancers, 2023, doi:10.3390/cancers15030911_

Round 1

Reviewer 1 Report

The authors describe an interesting approach to differentiate between endometrial cancer and benign causes for abnormal uterine bleeding using targeted protein analysis in cervical fluid instead of invasive diagnostic measures. The manuscript is well-written, easy to follow and includes figures and tables that add to the understanding of the project. 

It is well described how the two best suitable sampling methods were chosen and the detection rate of proteins. However, the inter-individual variability in protein detection rate seems to be rather big. A better description and, most of all, a discussion about this is warranted. Could the detection rate be optimized?

In the methods section, it should be stated when the samples are collected (years).  It would also be better to separate the endomtrial cancer cases in low and high grade, in line with current international guidelines, instead of grades 1-3. 

My main objections regard the lack of clinical reasoning. Since the aim of the project eventually is to introduce a new diagnostic tool, the manuscript would benefit from a description and correlation between the analyzed proteins and the known clinical features of the malignant cases. In the current manuscript, nothing is said about the discriminating function of the analyzed proteins in a clinical context . Eg., did the protein analyses detect all malignant cases equally well, or was there an overrepresentation of high grade and/or high stage tumors? Endometrioid vs. non-enodmetriod histologies? Was there a correlation or association with myometrial invasion or lymphovascular invasion? Were any other biomarkers in the malignant cases, like MMR-deficiency, analyzed. And, what were the causes of non-malignant uterine bleeding? 

Ie., the authors have to discuss their results in a potential clinical context too, and not only from a laboratory, diagnostic point of view. 

Reviewer 2 Report

In the manuscript entitled ” Cervical fluids are a source of protein biomarkers for early non-invasive endometrial cancer diagnosis” by Martinez-Garcia et al, is a study comparing ways of sampling cervico-vaginal fluids to be used in proteomic based studies. The endpoints protein quantity and proteome composition was used. For this part of the study four women seeking for abnormal uterine bleeding is included. They conclude that brush M1 and M3 gave the highest protein yield and M1 gave lowest variability.

The intention of this part of the study is very good but the study design give rise to several questions.

1.     Why so few patients? Results become hard to interpret and comparisons impossible. For example, how can you discuss variability with 2 patients in the group (line 232-233)? Please don´t highlight this results strongly.

2.     Along with above, in fig 2F it is not possible to suggest a common clustering for EC vs non-EC patients when you have only 2 samples clustering (EC) and 2 that do not (non-EC). Please change and address this.

3.     When was the samples collected in relation to final diagnose (EC/non-EC). Days, weeks, months before? Was the patient in anesthesia or awake when sampled?

4.     What was the diagnoses of the non-EC women?

5.     How was the diagnose reach for both EC and non-EC. Did all have surgery?

6.     Was the samples collected in consecutive order in the same patient. I.e Patient 1 brush M1, then M2, then, M3 and so on. You would think there is less cells/smear and proteins to collect when you come to the fifth brush.

7.     Was the order of brushes used the same for each patient?

8.     With regard to question 6 and 7, could the sampling order have affected the MS-analysis?

9.     How much protein (equal amounts/sample) was used in the proteomic analysis?

10.  Why different amount of PBS in the tubes?

In the second phase of the study, 52 proteins discovered in a prior study, was validated in all samples from the different sampling techniques in the 4 women. Again the M1, M3 brushes detected most of the proteins and a high FC between EC and non-EC was noted. Again the authors have to be careful with the interpretation of data due to the low number of included samples. I would even stretch to say that figure 3C I misleading since it is only describing random data.

Now the 2 best brushes M1and M3 was used to collect cervicovaginal fluid + UF from 41 women presenting with uterine abnormal bleeding and the 52 proteins were tested on these samples.

1.     Have this cohort of 41 women been used in any prior of the authors studies?

2.     When was the samples collected in relation to final diagnose (EC/non-EC). Days, weeks, months before? Was the patient in anesthesia or awake when sampled?

3.     What was the diagnoses of the non-EC women?

4.     How was the diagnose reach for both EC and non-EC. Did all have surgery?

Discussion

1.     In the 41 validation group UF and CF proteins separating EC and non-EC are not impressingly overlapping. Is CF not good enough for this type of sampling? Should we stick with UF in the clinic? Please discuss

2.     And the levels of proteins greatly differ between samples obtained from different sites. How can you get better explore the preanalytic technical conditions? Or is it patients that differ too much for this type of small discovery study. Please discuss.

Reference 21 och 4 are the same reference. Please check your references.

This study has very good intentions and set up, and addresses important aspects of new diagnostic tools for cancer diagnostics, that is often overlooked. I mean especially the preanalytic sampling workflow, when to sample, how, who is included, with what device, collected in which vial, transported how and so on. This paper take on these important part that is often not addressed when focusing on the MS/MS technique and ROC curves in the discovery process. Even though it raises new questions.

Reviewer 3 Report

It seems that in the introduction there is not enough information about the possibilities of diagnosing endometrial cancer in a minimally invasive way.The authors did not cite the possibility of determining HE4 or serpins due to serum. In addition ,endometrial biopsy seems to be a minimally invasive procedure that gives the histopathology examination and is almost 100% reliable. the authors ahould focus on talking about the selection of high risk group patient with these markers. Moreover ,as is the case in science not to consider that these initial studies provide the opportunity to formulate such conclusions with which this manuscript ends.

Reviewer 4 Report

In this article, Authors evaluate the potential of exo- and/or endo-cervical samples collected with 73 five different sampling methods to identify protein biomarkers for endometrial cancer (EC) detection. This article represents a valid contribution to the field, since this method seems to have the potential to become an useful item in the diagnosis of EC. 

I have the following comments to the Authors:

·      Introduction: Authors wrote that “Biopsies are preferably obtained by aspiration from the uterine 61 cavity (i.e., pipelle biopsy) as this is a minimally invasive sampling method”. In fact, this diagnostic method to obtain biopsies of a suspected EC is not very effective, having a sensitivity of about 47%, as Authors wrote. Hysteroscopic biopsies represent the gold standard in the diagnosis of EC, since hysteroscopy allows the operator to see the neoplastic tissue in the uterine cavity, as so should be suggested as the preferable method.

·      Methods: In order to make the study reproducible for other researchers, Authors should be more specific about the description of inclusion and exclusion criteria. Moreover, Authors should describe how was selection bias excluded during this phase.

·      Methods: Authors should clarify how were diagnosed the EC cases. Were they diagnosed by pipelle biopsy? This may represent a limitation of the study for the validation of the accuracy of cervical samples in the diagnosis of EC, since pipelle biopsy is a low sensitivity method and should be not suggested as the diagnostic gold standard.

  • Discussion: Metabolomics has recently appeared as a promising test for a non-invasive diagnosis of several diseases and metabolites were found able to predict the presence of EC tumor behavior (progression and recurrence) and pathological characteristics (histotype, myometrial invasion and lymph vascular space invasion). Authors may include in the discussion a brief revision of these novel circulating predictors (e.g. PMID: 32180221; PMID: 36139068), which could have an extraordinary impact on the management of EC in the future.

Round 2

Reviewer 3 Report

The current form of the manuscript is acceptable